# The Applications of Lattice Light-Sheet Microscopy for Functional Volumetric Imaging of Hippocampal Neurons in a Three-Dimensional Culture System

**DOI:** 10.3390/mi10090599

**Published:** 2019-09-11

**Authors:** Chin-Yi Chen, Yen-Ting Liu, Chieh-Han Lu, Po-Yi Lee, Yun-Chi Tsai, Jyun-Sian Wu, Peilin Chen, Bi-Chang Chen

**Affiliations:** Research Center for Applied Sciences, Academia Sinica, Taipei 11529, Taiwan

**Keywords:** lattice light-sheet microscopy, 3D cell culture system, functional neuron imaging

## Abstract

The characterization of individual cells in three-dimensions (3D) with very high spatiotemporal resolution is crucial for the development of organs-on-chips, in which 3D cell cultures are integrated with microfluidic systems. In this study, we report the applications of lattice light-sheet microscopy (LLSM) for monitoring neuronal activity in three-dimensional cell culture. We first established a 3D environment for culturing primary hippocampal neurons by applying a scaffold-based 3D tissue engineering technique. Fully differentiated and mature hippocampal neurons were observed in our system. With LLSM, we were able to monitor the behavior of individual cells in a 3D cell culture, which was very difficult under a conventional microscope due to strong light scattering from thick samples. We demonstrated that our system could study the membrane voltage and intracellular calcium dynamics at subcellular resolution in 3D under both chemical and electrical stimulation. From the volumetric images, it was found that the voltage indicators mainly resided in the cytosol instead of the membrane, which cannot be distinguished using conventional microscopy. Neuronal volumetric images were sheet scanned along the axial direction and recorded at a laser exposure of 6 ms, which covered an area up to 4800 μm^2^, with an image pixel size of 0.102 μm. When we analyzed the time-lapse volumetric images, we could quantify the voltage responses in different neurites in 3D extensions.

## 1. Introduction

In recent years, the rapid development of microfabrication techniques has enabled us to study cellular behavior in a well-controlled microenvironment, mimicking the native environment of the disease states [1,2,3,4]. With the integration of microfluidic systems and three-dimensional (3D) cell cultures, organs-on-chips have shown great potential in high throughput drug screening applications [5,6,7,8]. In order to faithfully reconstruct the in vivo microenvironment, 3D culture systems are often used for the organs-on-chips, which raises a challenging issue in detecting the response of individual cells with high spatiotemporal resolution while minimizing the damage of cells during the observation process. In the conventional approach, microfluidic devices are placed on an inverted microscope, where the scattering from multiple layers of cells hampers light penetration, thus leading to low imaging quality. Therefore, one of the key issues in the development of the organs-on-chips system is to monitor the spatiotemporal behavior of individual cells in the 3D cell culture.

One of the major advantages of the organs-on-chips is the capability to mimic diseases at the organ level on chips. Among various diseases, neurodegenerative diseases are of great research importance because these diseases are currently considered incurable [9]. Two-dimensional (2D) neuron cultures are the most studied systems for understanding neurodegenerative diseases, in which neuronal behavior can be manipulated and measured through various techniques [10,11,12,13,14]. The function of neurons can be monitored through membrane potential or calcium dynamics using optical microscopic tools [15,16,17,18]. In general, the application of calcium indicators has yielded satisfactory results for primary 2D cultures, slices of brain tissue, and in vivo brain imaging [19]. However, it remains difficult to measure neuronal activity in 3D due to the limited light penetration and imaging speed. At present, high-speed calcium imaging from 2D or 3D cultured neurons can be recorded at a fixed focal plane [20]. Alternatively, multi-point scanning spinning disc microscopy [21] also offers high speed imaging with low phototoxicity. However, the penetration depth of light in 3D culture is always problematic, which may be partially resolved by the use of two-photon microscopy [8,22,23]. However, recording neuronal activity in 3D cultures requires the development of high-speed volumetric imaging techniques. Recently, selective plane illumination-based techniques have been demonstrated to be capable of high-speed volumetric imaging, [24,25]. Among them, lattice light-sheet microscopy (LLSM) has been shown to offer several advantages over other volumetric imaging tools, including less photobleaching and phototoxicity and better subcellular imaging resolution [26,27]. In LLSM, a pair of microscope objectives with perpendicular orientation shares the same focal point, where the sample is placed. The specimen and both tapered ends of the objective are immersed in the medium filled, temperature-controlled chamber. A 2D optical lattice composed of hundreds of Bessel beams scans across the whole sample and generates a 3D fluorescent image with resolutions of 250 nm laterally and 500 nm axially, at a speed of several milliseconds per excitation plane. Therefore, in this study, we utilized LLSM to monitor neuronal activity via subcellular imaging of voltage and calcium dynamics. 

## 2. Materials and Methods 

### 2.1. Preparation of Rat Hippocampal Neurons for 3D Culture

Animal experiments were conducted under the guidelines of the Academia Sinica Institutional Animal Care. Postnatal day 0 (P0) rats were sacrificed, and hippocampal tissues were dissected as described in previous experiments [28,29]. Hippocampal neurons, isolated by papain-mediated isolation, were kept in ice-cold medium, mixed with Matrigel (Corning) in a 1:1 ratio [30], and seeded on 35 mm glass-bottom dishes (MatTek Corporation, Massachusetts, MA, USA) (Figure 1). The mixture of Matrigel and neurons was added in the center of a MatTek dish. An 18 mm coverslip was then placed on top of the mixture droplet, resulting in a spread round sheet with a thickness of ~2 mm. A consistent size of 3D culture gel could be realized by this approach. The mixture of Matrigel and neurons would rapidly become solidified at 22 °C to 35 °C. Those procedures were performed on ice to prevent gel polymerization. The neurons were allowed to form a round sheet of 3D gel at a density of 10^4^/mm^3^ at 37 °C for 2 h. After gelation, 2 mL of neuronal medium were added into each dish. Hippocampal neurons were maintained in neuron culture medium (Neurobasal-A medium, Thermo Fisher Scientific, Massachusetts, MA, USA), supplemented with GlutaMAX (Thermo Fisher Scientific, Massachusetts, MA, USA), B-27 (Thermo Fisher Scientific, Massachusetts, MA, USA), and 20% glial conditional medium, and incubated at 37 °C in a 5% CO_2_ environment. Glial cells were cultured only for collecting their medium, which have secreted growth factors and some unidentified factors for providing neuron health during culture.

### 2.2. Immunofluorescence Staining

In order to verify the condition of cultured neurons, we conducted neuronal immunostaining using various markers. The neurons were rinsed with ice-cold PBS-MC (phosphate-buffered saline, Sigma Aldrich, Missouri, MO, USA) with 1 mM MgCl_2_ and 0.1 mM CaCl_2_, and fixed with 4% paraformaldehyde (PFA) and 4% formaldehyde (paraformaldehyde aqueous solution; EM Grade, Electron Microscopy Sciences/4% sucrose/PBS-MC) for 20 min at room temperature (RT). Neurons were rinsed three times with PBS-MC and incubated with blocking buffer (4% goat serum (Gibco)/2% bovine serum albumin (BSA; Sigma Aldrich, Missouri, MO, USA)/PBS-MC) for 30 min at RT. Before antibody staining, neurons were permeabilized with blocking solution containing 0.2% TritonX-100 (Sigma Aldrich, Missouri, MO, USA) for 30 min at RT. The neurons were rinsed three times with PBS-MC and then stained with primary antibodies, such as anti-Tau-1 (1:500; Millipore, Burlington, Massachusetts, MA, USA), postsynaptic density protein-95 (PSD-95) (1:200; Millipore, Burlington, MA, USA), synaptotagmin-1 (1:200; Synaptic Systems, Beijing, China), and microtubule-associated protein-2 (1:200; MAP2; Chemicon, Osaki, Tokyo, Japan), which were diluted in blocking buffer and incubated at 4 °C overnight in a humid chamber. Neurons were rinsed three times with PBS-MC at RT. In the final stage, the secondary antibodies, goat anti-mouse-conjugated Alexa Fluor 488 and goat anti-rabbit-conjugated Alexa Fluor 546 (1:1000 for both antibodies; Jackson ImmunoResearch, Shanghai, China), were diluted in blocking buffer and incubated with neurons at 4 °C overnight in a humid chamber. After three rinses with PBS-MC at RT, neuron gels were kept within PBS-MC until imaging was acquired by an inverted research microscope system (Leica DMi8, Illinois, IL, USA) (Figure 1B). The neurons which were cultured on coverslips were mounted in Prolonged Gold antifade reagent (Invitrogen, Carlsbad, CA, USA) according to the manufacturer’s instruction with overnight curing in the dark at RT, and images were recorded with a confocal microscopy LSM880 (ZEISS, New York, NY, USA) (Figure 1A).

For Figure 1A, immunostained images were acquired in z-stacks with an upright laser scanning confocal microscopy LSM880 (ZEISS, New York, NY, USA) with a 63× (NA = 1.4) oil objective. Alexa Fluor 488 and Alexa Fluor 546 fluorescent dyes were excited with the 488 and 561 nm laser lines, and collected within 510–550 and 570–620 band filters, respectively. Each channel was acquired separately to minimize bleed-through. Images were processed by ZEN software (ZEISS, New York, NY, USA).

The two-color Nneuronal 3D gel images (Figure 1B) were recorded with a confocal microscopy LSM880 (ZEISS, New York, NY, USA) (Figure 1A) and an inverted research microscope system (Leica DMi8), Leica DMi8 microscope equipped with a 40×/1.3 NA oil-immersion objective, Colibri LED Illumination system, 450–490 nm and 540–580 nm band pass filters. Neuronal 3D gel images were captured by an electron multiplying CCD (Evolve 512, Photometrics, Arizona, AZ, USA) (Figure 1B) in 16-bit scale and were processed by ImageJ (NIH).

### 2.3. Setup of Lattice Light-Sheet Microscopy

The schematic diagram of our microscopic system is depicted in Figure 2. In order to record 3D neuronal activity, we constructed LLSM as described previously [26] with a slightly modifications of sample holder and stages, which were built on an inverted microscope (Olympus IX71, Olympus Corporation, Tokyo, Japan). Two excitation lasers (λ = 488 and 561 nm) were used in this experiment. The light-sheet was generated via either a low numerical aperture (NA) configuration (excitation objective (Nikon, 40× CFI APO NIR, 0.8 NA, Tokyo, Japan) and detection objective (Nikon, 40× CFI APO NIR, 0.8 NA)) or a high NA configuration (excitation objective (special optics, NA = 0.66) and detection objective (Nikon, CFI Apo LWD 25×, NA = 1.1)). The emission signals were imaged by an sCMOS (Hamamatsu, Orca Flash 4.0 v2 sCMOS) detector, (Hamamatsu, Iwata, Japan). The high NA microscope features near-isotropic resolution in all directions by very thin plane scanning to get the lateral and axial resolution, with 200 nm and 400 nm. By using a piezo scanner (Physik Instrumente, P-726 PIFOC, Karlsruhe, Germany), the movement of the detection objective can be synchronized with the excitation objective. For image acquisition, a 3D gel sheet sample was mounted on a regular microscope glass slide attached on a 3D translational stage. The gel sheet was then immersed in the imaging buffer (Hank’s balanced salt solution (HBSS) buffer) (Sigma Aldrich, Missouri, MO, USA) meniscus formed between the objectives and the glass slide.

### 2.4. Voltage and Calcium Dye Labeling for Functional Imaging

In order to record the neuronal activity, 4 μM Fluo-4AM (Thermo Fisher Scientific, Massachusetts, MA, USA) was added to the 14–19 days in vitro (DIV) neuronal culture for calcium imaging, whereas 0.1 μM di-4-ANEPPS (Thermo Fisher Scientific Massachusetts, MA, USA) was used for voltage imaging. During the experiment, neurons were supplied with Hank’s balanced salt solution (HBSS) buffer (Sigma Aldrich, Missouri, MO, USA) (137 mM NaCl, 5.4 mM KCl, 0.25 mM Na_2_HPO_4_, 0.44 mM KH_2_PO_4_, 1.3 mM CaCl_2_, 1.0 mM MgSO_4_, 1.0 mM MgCl_2_, 10 mM glucose, and 10 mM HEPES; pH 7.4) containing 0.01% Pluronic F127 (Thermo Fisher Scientific, Massachusetts, MA, USA) and 0.1% bovine serum albumin (Sigma Aldrich, Missouri, MO, USA) [31,32]. Calcium and voltage dyes were loaded at 37 °C under 5% CO_2_ environment for 50 min. 

The acquisition time for one 3D volume image was between 3 and 5 s All of the experiments were performed at room temperature. In order to obtain images, KCl-induced membrane potential and changes in calcium influx, images of 3D-cultured neurons were acquired for 1 min at baseline, in which no significant photobleaching was observed, and signals were averaged as F0.

### 2.5. Setup of Electrical Stimulation of Neurons

In order to electrically stimulate the neurons, two parallel electrodes were attached to the glass slide, with a separation distance of 1.5 cm. Electric pulses were generated through a data acquisition device (DAQ, USB-6341, National Instruments, Texas, TX, USA), in which the analog output was connected to one of the scaling amplifiers (SIM-983 in SIM-900 mainframe, Stanford Research Systems, California, CA, USA). The output of the amplifier was connected to the electrodes on the glass slide using a 50 cm off-the-shelf BNC-to-alligator cable (RG-58/U, Jun-Mao, Taipei, Taiwan). Electrodes ran parallel to the long side of the slide in order to avoid mechanical interference during acquisition. In order to calibrate the strength of the electrical field with respect to the provided voltage level, a secondary set of paraxial electrodes was placed in between the first set in order to measure the sensed voltage. Temporal parameters of the stimuli, including amplitude, pulse interval, and duration, were controlled through a customized LabVIEW program. During electrical stimulation, electric pulses with a magnitude of 10 V/cm at 10 Hz were applied to the samples for a total of 900 pulses through an amplifier (Stanford Research Systems, SR830, California, CA, USA).

### 2.6. Image Processing and Analysis

In order to quantify the voltage responses, the maximum intensity at each time point was projected onto one 3D image and used for the selection of region of interest (ROI) in 3D. This 3D projection image of the time-lapse image was derived from registered images with maximum intensity projection over time, which served as the entire neuronal contour. The ROIs were depicted by hand drawn profiles for each z-stack. The label analysis function of Amira was used to calculate the values in 3D ROI. 

Segmentations of neurites and soma were automatically traced using the XTracing pack of Amira (Thermo Fisher Scientific, Massachusetts, MA, USA) (Weber et al., 2012; Rigort et al., 2012). We followed the protocol of the Amira User Guide [33] using the parameters of the cylinder correlation module as listed below (μm): (1) cylinder length of 63.7; (2) angular sampling of 5; (3) mask cylinder radius of 14; (4) outer cylinder radius of 12; and (5) inner cylinder radius of 0. The color codes of each neurite and soma were selected independently. 

The functional images of 3D neurons were processed by ImageJ (NIH) and Amira 3D Software (Thermo Fisher Scientific, Massachusetts, MA, USA). Prism (Graph Pad) and Excel (Microsoft Office) were used for data analysis. For quantitative analysis, the initial fluorescence Ca^2+^ and membrane potential intensities within 1 min were averaged and used as the baseline fluorescence intensity (F0). The fluorescence intensity of each time point was defined as Ft. For the fluorescence change (ΔF) before and after stimulation, we calculated the change by the equation ΔF = ([Ft−F0]/F0) (%).

## 3. Results

### 3.1. Characterization of Hippocampal Neurons in a 3D Culture System

There are various types of 3D culture systems for the study of neuronal development and differentiation [34,35,36]. Based on previous reports [30], we developed a modified protocol for a 3D culture system to observe the differentiation and maturation of primary hippocampal neurons. For primary hippocampal neurons, it takes approximately 15 days to develop complete neuronal networks and synapses. To establish the 3D environment for neurons, the neurons were cultured within 50% Matrigel [30] under neuronal growth and differentiation medium, as described in a regular culture system [28]. The thickness of the Matrigel was controlled by laminating a coverslip with a Matrigel droplet on a glass-bottom dish, which was around 2 mm. From the bright field z-stack images (approximately 250–300 μm in total depth) of 3D-cultured neurons (Appendix A with field of view at 400 μm × 400 μm × 160 μm, z step size of 1.6 μm), it was found that the neurons were evenly distributed within the 3D environment. This result suggests that our modified protocol could successfully create a 3D environment for neurons.

After 10 days in vitro (DIV) culture, synaptogenesis started in neurons and formed synapses between neurons for transmission at the neuronal network. A mature network was formed after 15 DIV. In order to compare hippocampal neurons cultured in 2D and 3D environments, we examined axon and dendrite formation by staining Tau and MAP2 proteins at an early stage (4 DIV), and the formation of synapses using synaptotagmin to label presynaptic terminals and PSD-95 for postsynaptic terminal staining at a later stage (15 DIV) (Figure 1). Our experimental results indicate that the 3D culture neurons could successfully differentiate and form a neuronal network, suggesting that this modified protocol can provide an environment for neuronal differentiation and maturation in a 3D culture system similar to those neurons in organoid or tissue cultures. Unlike the neurons grown in the 2D culture system, the neurons cultured in the 3D system could extend their neurites over 150 μm in the z direction, which is ~six-fold longer than those found in 2D cultured neurons (20–25 μm). Therefore, we used this modified 3D culture system to mimic the neuronal spheres or brain organoids grown on chips [36,37].

### 3.2. Lattice Light-Sheet Microscopy (LLSM) for 3D-Cultured Neuron

To visualize the neurons in the 3D culture system by conventional fluorescence microscopy, strong scattering from thick samples leads to blurred fluorescence signals (Appendix A). The limitation of scattering and light penetration can be improved by the use of light-sheet excitation, which offers several advantages for monitoring the neuronal activity in a 3D culture system, including a reduction in photobleaching, phototoxicity, and scattering [26,27,38]. In the original LLSM setup, two orthogonally aligned water dipping objectives were immersed in an imaging medium bath with a volume of ~10 mL [26]. For the drug treatment or chemical stimulation experiments, this large liquid-filled chamber will become problematic, especially for pricing chemicals. In order to study the 3D-cultured hippocampal neurons, we modified the original design of the LLSM by replacing the liquid-filled chamber with a microscope glass slide (Figure 2A) mounted on an inverted microscope (Figure 2B). The combination of LLSM with an inverted microscope allows us to monitor the locations of samples through eyepieces (Figure 2B). For refraction index matching, approximately 1 mL of imaging buffer (HBSS) was added to the center of the microscope glass slide in order to create a water bridge between the two objectives (Figure 2C,D) [39]. The 3D-cultured hippocampal neurons can be imaged in a small volume of liquid compared with the original design, which requires about 10 to 12 mL of imaging buffer [26]. The reduced volume of imaging buffer is especially beneficial for pharmaceutical studies, in which reagents are limited and sometimes very expensive. 

In order to monitor neuronal activity, voltage probes for imaging the membrane potential as a record of neuronal activity of the 3D-cultured hippocampal neuronal activity are required. However, there are only limited probes options in the market. In this study, we utilized a commercially available voltage sensitive aminonaphthylethenylpyridinium dye, di-4-ANEPPS, to image the membrane potential, because of its high signal-to-noise ratio, positive correlation with voltage dynamic, and a red-shift emission [40,41]. In order to perform the live labeling of 3D-cultured neurons with voltage, we adapted previously published procedures [31,32]. The 3D spatial distribution of voltage response on the neuron membrane can be mapped out by the customized LLSM (Figure 2F and Appendix A), while different angles of views of a neuron can be seen in Figure 2G. These results revealed that the neurite outgrowths of the primary hippocampal neurons cultured on the Matrigel-based system were extended in 3D. In addition, we demonstrated that the LLSM is capable of recording three-dimensional voltage responses of the primary neuron culture. Indeed, image shown in Figure 2F exhibits significantly improved imaging quality when compared with those images taken by conventional microscope.

### 3.3. Quantification of Voltage Responses at Different Subcellular Areas

Quantification and analysis of neuronal function and morphology is an active research area that has been ongoing for decades because of its importance in our understanding of neuronal degenerative diseases [42,43,44]. In order to monitor neuronal activity, we utilized LLSM to record the three-dimensional voltage response of hippocampal neurons on a 3D culture system stimulated by 25 mM of KCl solution. The selected images at different time points are displayed in Figure 3A, and the entire time-lapse image can be found in the supporting information (Appendix A). The color-coded intensity indicates the change in fluorescent intensity throughout the acquisition period before and after the KCl treatment. Since LLSM provides the 3D voltage response information in neuronal cells, it is very important to develop a 3D quantitative analysis method in order to quantify the voltage response in 3D. We first conducted neuronal segmentation using the z-stack images, in which five neurites and soma were identified. In order to calculate intensity changes in each neurite over time, we selected a region of interest (ROI) in each neurite and soma. The segmented neuron is shown in Figure 3B(a)and the ROIs are displayed in Figure 3B(b), in which the overlay image is shown in Figure 3B(c) (Appendix A). Using the integrated intensity in the ROIs, we compared the change in fluorescence intensity at different locations and time points. In the initial stage, the absolute fluorescence intensity of soma was higher than those observed in the neurites (Figure 3C). After KCl stimulation, the fluorescence intensity of soma increased at a faster rate compared with that of neurites (Figure 3A). In order to quantify the voltage response at different times, we calculated the changes in integrated intensity (ΔF/F0 (%)) at different ROIs using the integrated fluorescence intensity of each ROI at time zero as a reference (F0) (Figure 3D). Changes in fluorescence intensity were observed for each neurite after KCl stimulation, in which fluorescence intensity was found to increase in three out of five neurites. 

During image analysis, we found that di-4-ANEPPS dyes labeled not only the plasma membrane but also the cytoplasm of a soma. We further used a 3D graphic software to visualize the intensity change in soma after KCl treatment (Figure 3E). The orthoslice images of the voltage response show that the area with maximum intensity was adjacent to the nucleus instead of the plasma membrane. This result suggests that the voltage dye, di-4-ANEPPS, may not only anchor on the membrane but also accumulate in the cytosol, which could not be resolved by conventional microscopic tools, indicating the superior performance of LLSM for 3D cell cultures.

### 3.4. Calcium Imaging of Neuronal Activity under External Stimulations

Calcium ions are another common indicator for measuring neuron activity. In a typical cell, the intracellular free calcium concentration is about 50–100 nM, which is 104 times lower than the extracellular concentration [45]. In order to monitor changes in calcium concentration, fluorescence dyes, such as Fluo-4AM, are often used [19,46]. In neuronal cultures, the neuronal network would release neuronal transmitters, which results in spontaneous neuronal firing. In this experiment, we employed LLSM to record the spontaneous activity of 3D-cultured hippocampal neurons labeled with Fluo-4AM [47]. We observed low-frequency spontaneous spiking in our 3D culture system (Figure 4A). In addition to spontaneous firing, neuronal activity can also be stimulated by chemicals or electric pulses. We first stimulated the 3D-cultured hippocampal neurons with 25 mM KCl (Figure 4A, Appendix A). A significant increase in the fluorescence intensity of Fluo-4-AM was observed, indicating functional activity and calcium influx in 3D-cultured hippocampal neurons (Appendix A). In order to test the behavior of the 3D-cultured neurons under electric pulse stimulation, two electrodes were placed in parallel on the microscope glass slide, and an electric field of 10 V/cm was applied to the electrodes at 10 Hz. We observed wave-like calcium responses [48] (Figure 4B and Appendix A), suggesting that the 3D-cultured neurons could form a functional neuronal network and the neuronal activity could be stimulated by both chemicals and electric pulses. 

In order to quantify the spontaneous neuronal firing, we analyzed the fluorescence intensity change of neurons in the selected ROIs (Appendix A). Appendix A shows the maximum intensity projection image of calcium response in neurons, in which volumetric images were acquired at a rate of 1 second per volume. Figure 4C shows the changes in mean fluorescence intensity. In this experiment, spontaneous firing behavior was observed in 27.3% of neurons in 3D culture (3 out of 11 neurons from two independent experiments). The calcium response can also be found in neurons under chemical stimulation (KCl). Different temporal behaviors were observed in different neurons (Figure 4D). The fluorescence intensities of four neurons (Appendix A) were found to increase immediately after stimulation and decay with different behavior. Whereas neuron 3 exhibited a single decay, bimodal responses were found in neurons 1 and 2. 

In order to further investigate neuronal activity under electrical stimulation, we applied an electrical field of 10 V/cm between two paralleled electrodes on the slide [49], which were generated by scaling amplifiers using customized software. After acquiring the baseline images, the neurons were stimulated with a sequence of 900 electric pulses with 1 msec pulse width at 10 Hz. The electrical pulse stimulation can induce the release of neuronal transmitters, resulting in neuronal activation. By applying the electric field to the 3D-cultured neurons, we could monitor the synaptic activity of neurons via analysis of calcium influx of spines and boutons (Figure 4E). The intensities of calcium were quantified through the selected ROIs (Appendix A). Some (but not all) of the boutons also presented a similar behavior after several hundred rounds of electric pulse stimulation. These results indicate that this 3D-cultured neuron system could provide a functional network, and the synaptic plasticity of spine and boutons in the 3D-cultured system could be observed and analyzed by LLSM.

## 4. Discussion

Electrophysiological techniques are currently regarded as the most accurate approach for providing membrane potential information in neurons, since this approach directly measures and records the conductance of a single neuron [50]. However, this approach provides limited information on the directional and spatiotemporal dynamics of the voltage change. In addition, such manual operation is difficult to scale up, especially for on-chip screening applications. In this experiment, we demonstrated that LLSM could be used as an alternative tool for measuring neuronal activity with very high spatial resolution in 3D culture systems. Overcoming the limitations of conventional fluorescence microscopy, LLSM has presented great potential in the study of functional aspects of hippocampal neurons in a 3D environment. It provides excellent optical properties for time-lapse imaging, allows parallelization for large-scale studies, and reveals detailed subcellular information with low photobleaching and phototoxicity effects [38]. In this study, the 3D Matrigel-embedded hippocampal neurons could differentiate and mature as regular 2D culture (Figure 1). Figure 1A shows the confocal images of 2D fixed cultured neurons with good spatial resolution because of point-scanning with a placed pinhole, while Figure 1B shows the images of 3D cultured neurons taken by wide-field microscopy with worse 3D image quality due to the fluorescent background from the thick sample. In order to record the chemical or electrical response of 3D cultured neurons at high spatiotemporal resolution, we built LLSM to perform such experiments for its good 3D imaging capability, such as optical sectioning and live imaging. The 3D images acquired by LLSM demonstrate that the hippocampal neurons could elongate the neurite outgrowth toward different orientations (Figure 2F,G). Moreover, the inset of Figure 2F shows the photobleaching curve of the observed samples, there was no obvious bleaching happening over our experimental time. LLSM was used to observe the functional dynamics of 3D-cultured hippocampal neurons (Figure 3 and Figure 4), which is the most fundamental requirement for visualized neuron activity. The chemically-induced increase in voltage was observed with a subcellular resolution by monitoring the intensity dynamic of di-4-ANEPPS in 3D (Figure 3) or with Fluo-4AM (Figure 4), while the spontaneous firing of hippocampal neurons monitored by a calcium indicator was captured in a large imaging field (Figure 4A,C). The calcium influx spikes of presynaptic terminal and postsynaptic spines could be observed by LLSM after the neurons were stimulated with electric pulses (Figure 4B,E). These results demonstrate the possibility of live 3D culture imaging, in which LLSM can serve as a simple and powerful approach for the study of neuronal activity in a 3D culture environment.

Although LLSM serves as excellent approach for functional imaging in the neuroscience field, there are still some limitations in the selection of fluorescent indicators. The major technical challenges of voltage imaging are the limited selection of voltage sensitive dyes, genetically encoded voltage indicators, and rapid firing events of action potential, which take sub-milliseconds to occur [16]. Even though we are unable to increase the temporal resolution of the voltage image as fast as electrophysiological changes occur, we could still progressively improve the spatial resolution for observing the dynamics of the subcellular region, which allows for direct analysis of the neuronal response from the soma and axon to dendrites (Figure 3 and Figure 4). Voltage-sensitive dyes were loaded into plasma membrane via the internalization by the neurons [51,52], which may have led to the observation that the di-4-ANEPPS dye accumulated in the cytosol and nuclear regions [53] (Figure 3E). Similar results were observed in a previous study, in which the voltage-sensitive dye, the fluorescent imaging plate reader membrane potential dye, FMP, behaved as a charged molecule and accumulated in the cytosol of soma [54]. 

According to our voltage and calcium neuronal images, these live imaging results revealed that not all neurons or subcellular structures respond in the same way during chemical stimulation (Figure 3D and Figure 4D). In this study, we used di-4-ANEPPS as a voltage dye, which was excited by a 488 nm laser and had a red emission shift, while the same laser was used to excite the Fluo4-AM calcium dye. Emission of Fluo4-AM could be separated by different band-pass filters. Therefore, the combination of voltage (di-4-ANEPPS) and calcium (Fluo4-AM) dyes with the same excitation but disparate emission wavelengths within the same neuron could provide advanced information about the neuronal connection and neuronal activity within a 3D network [32,47]. 

Although a two-photon imaging system could monitor the calcium dynamic in 3D-cultured systems or tissues [55,56], the images are restricted to a single focal plane, which loses some spatial neuronal information. Here, we used the LLSM to image the whole live 3D z-stack image and then analyzed the synaptic plasticity in a 3D-cultured system. Our results might open the road toward live functional imaging of 3D-cultured systems and show the possibility of imaging specimen in a 3D controlled microenvironment at high spatial and temporal resolution by LLSM. However, the throughput of preset microscopes will be the main drawback. The microfluidics can contribute to developing a high-throughput LLSM microscope. If LLSM is built based on microfluidics, the 3D controlled microenvironment for biological research could be studied easily and extensively for live imaging. Therefore, with the powerful 3D culture ability, microfluidics can be incorporated with advanced LLSM to develop a new platform for biology and biomedical research fields. Instead of PDMS, used mostly in microfluids, integrating these two systems with a water refractive index matching environment will be of interest for us. 

## 5. Conclusions

Cell populations in diseased tissues are very heterogeneous and extend over a 3D space, which is very different behavior than that seen in cells in 2D culture dishes. Therefore, in order to improve the efficiency in drug screening applications, organs-on-chips, which integrate 3D cultures with microfluidic systems, may provide more reliable results, because they better represent the native microenvironment within tissues. However, some cells in the 3D cultures are buried deep inside the cell mass, which cannot be easily accessed by conventional microscopic tools. In this study, we demonstrated that individual neurons in 3D culture can be visualized by LLSM. The volumetric images of neurons were done by scanning the light-sheet plane along the axial direction with frame exposure of 6 ms, which covered an area up to 60 μm × 80 μm with an image pixel size of 0.102 μm. With the voltage probe di-4-ANEPPS, 3D membrane voltage responses of neurons under chemical stimulation were monitored at different time points with subcellular resolution. It was noticed through the volumetric images of neurons that the voltage probes accumulated near the nucleus instead of the membrane, which cannot be resolved by conventional fluorescence microscopy. Calcium influx in neurons under external stimuli, such as chemical and electric stimulation, was investigated by LLSM, in which spontaneous spiking and wave-like behaviors were observed for neurons in 3D culture. In summary, LLSM can be used as a platform for the characterization of organs-on-chips with several advantages, including volumetric imaging, high spatiotemporal resolution, reduced photo-toxicity and photo-bleaching, deeper penetration, and the capability of recoding functional imaging in 3D.

## Figures and Tables

**Figure 1 micromachines-10-00599-f001:**
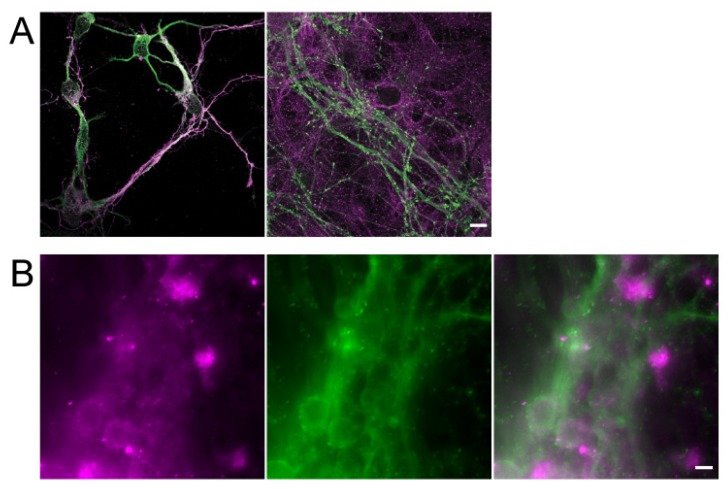
The characterization and comparison between 2D and 3D cultured hippocampal neurons. (**A**) Images of neurons cultured on 2D coverslips. (Left) early stage markers, Tau (magenta) and microtubule-associated protein 2 (MAP2) (green) are used to labeled axon and dendrites, respectively, at 4 days in vitro (DIV). (Right) mature neurons at 15 DIV were labeled with markers for synapse formation, Synaptotagmin 1 (magenta) for presynaptic terminals and postsynaptic density protein 95 (PSD-95) (green) for postsynaptic. Scale bar: 10 μm. (**B**) The differentiation of 3D cultured neurons was visualized by staining with the neuronal markers, Tau (magenta) and MAP2 (green) at 15 DIV. Scale bar: 10 μm.

**Figure 2 micromachines-10-00599-f002:**
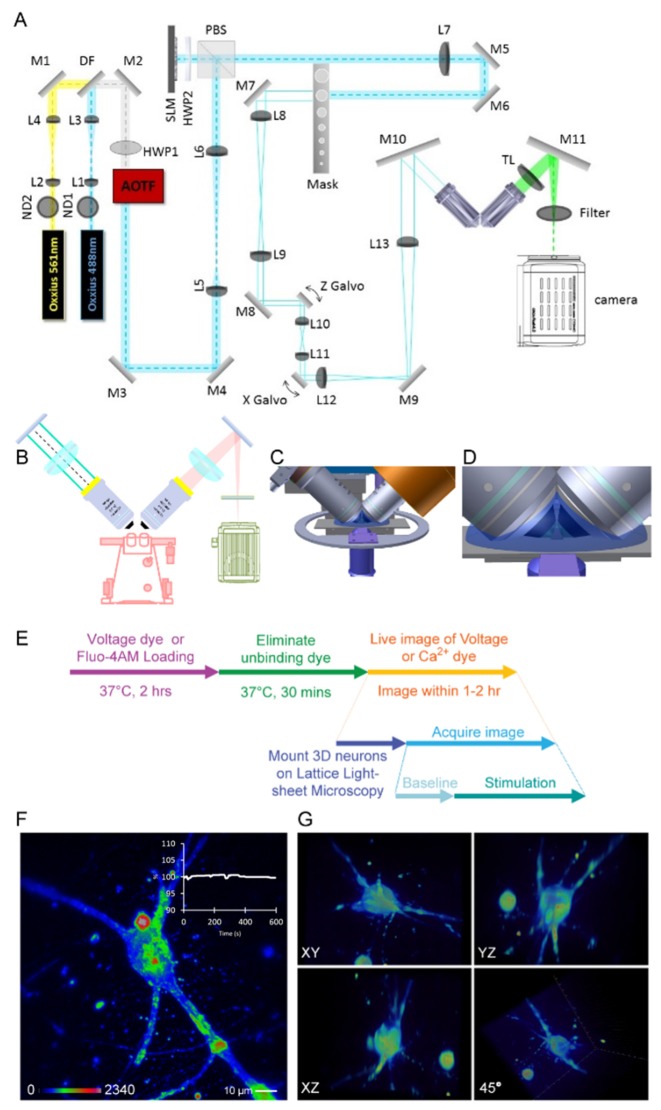
(**A**) The detailed optical schematic for the lattice light-sheet microscopy (LLSM) system. Abbreviations: L—lens, M—mirror, DF—dichroic filter, ND—neutral density filter, HWP—half-wave plate, AOTF—acousto–optical tunable filter, PBS—polarization beam cube, TL—tube lens. The system was built on an inverted microscope with two excitation lasers. (**B**) The excitation and detection objectives were mounted on an inverted microscope perpendicular to each other. (**C**) The design of the 3D culture sample holder and relative position of the orthogonal excitation (left) and detection (right) objectives. Samples were mounted on a slide, which was connected to a customized sample holder on the sample stage. (**D**) The sample was located in between two objectives and the space between objectives was filled with 1 mL imaging buffer. (**E**) The flowchart of voltage and calcium dye labeling procedures of 3D cultured neurons. (**F**) The maximum intensity projection of a live image of 3D hippocampal neuron labeled by voltage dye at T0. Inset shows the fluorescent intensity change of the 3D volumes over time. Scale bar: 10 μm. (**G**) Different angles of views of the 3D culture neuron shown in (F).

**Figure 3 micromachines-10-00599-f003:**
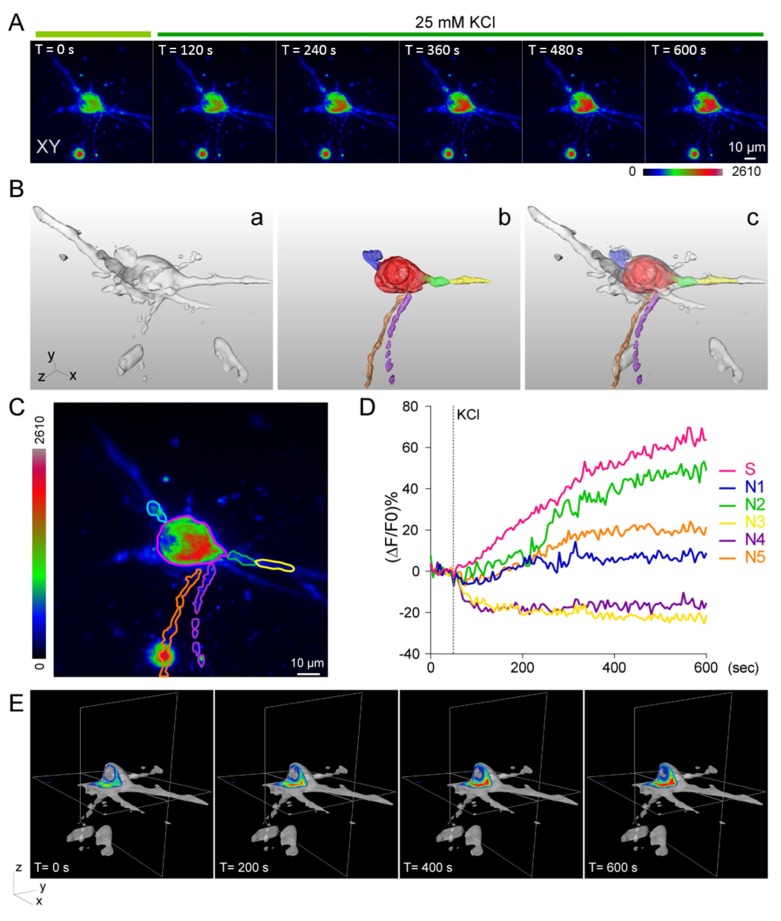
Voltage response of neuron to chemical stimulation (KCl). (**A**) Images of voltage dye-labeled neuron before and after 25 mM KCl treatment at different time points. Scale bar: 10 μm. (**B**) The 3D segmentation and ROI images of voltage dye labeling neurons. The T0 image of (A) was used for 3D reconstruction and segmentation. Selected 3D ROIs are shown in different colors. The relative positions between ROIs (a) and (b) are overlaid, as shown in (c). (**C**) The maximum intensity projection of the T0 image and the outlines of ROIs. (**D**) The changes in the integrated intensity in different ROIs at different times before and after 25 mM KCl stimulation. The dashed line indicates the time point of stimulation. The intensity is normalized with ROI volume. (**E**) Cross-sectional view of voltage dye at different time points.

**Figure 4 micromachines-10-00599-f004:**
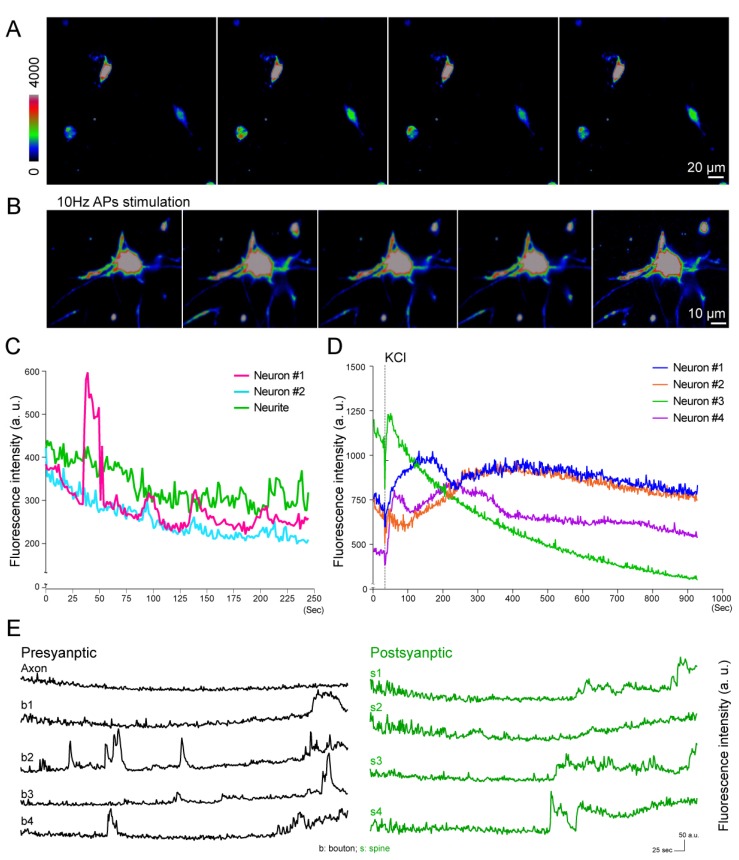
The calcium images of neuronal activity of 3D-cultured neurons. (**A**) Maximum projection intensity images (XY plane) of the spontaneous calcium activity of 3D-cultured neurons at different time points. Scale bar: 20 μm. (**B**) Maximum projection intensity images (XY plane) of 3D-cultured neurons under 10 Hz electric stimulation. Scale bar: 10 μm. (**C**) The fluorescence intensity of the calcium influx of neurite and somas from selected neurons in image S1 of supporting information. (**D**) The fluorescence intensity of calcium influx of selected neurons in image S2 of supporting information. (**E**) The fluorescent intensity dynamic of calcium responses from selected spines and boutons of 3D cultured neuron in images S3 of supporting information. Scale bars: 25 s and 50 a.u. (arbitrary units); b: bouton; s: spine.

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
