# Peer review of "The Applications of Lattice Light-Sheet Microscopy for Functional Volumetric Imaging of Hippocampal Neurons in a Three-Dimensional Culture System"

_micromachines, 2019, doi:10.3390/mi10090599_

Round 1

Reviewer 1 Report

Micromachines-582536: The Applications of Lattice Lightsheet Microscopy for Functional Volumetric Imaging of Hippocampal Neurons in a Three-Dimensional Culture System

General remarks

In this paper, the authors aim at demonstrating the suitability of LLSM to image the stimulation of hippocampal neurons embedded in Matrigel. They found that LLSM enable to precisely resolve, at the subcellular level, the calcium and voltage signaling after stimulation with KCl or electric pulse, in a time resolved manner. In addition, they found that LLSM enables to visualized that di-4-ANEPPS resided mainly within the cytosol instead of the membrane.

While this paper is interesting, several items need to be clarified, before considering the publication of the manuscript in Micromachines.

Specific remarks

Introduction
Page 1, lines 31-43, page 6 line 184, page 11 line 370: the authors discussed the advantages of organ-on-chip to better mimic organ behavior in vitro. It is not clear how the presented methodology can be translated to chip made with PDMS where the imaging working distance is significantly higher in comparison to glass slides.

Materials and methods
Page 3, lines 80-90. The authors should detail the immunostaining protocol: the dilution of each primary antibody, the time of incubation with each solution (i.e blocking, primary antibodies etc.), as well as the type of negative controls to ensure the specificity of the primary antibodies binding.

Page 5 line 158. It is not clear how many times were repeated the stimulation experiments with different neural populations (i.e. biological replicates).

Results and discussion

Page 5-6, lines 160-171: The authors should provide quantitative information about the purity of the neuronal population they isolated (i.e. the possible percentage of remaining astrocytes, that can also react to KCl stimulation).In addition, they should provide a quantification of the neural differentiation, as well as the percentage of differentiated neurons at the beginning of calcium and voltage experiments.

Page 7 line 219: the authors should clarify on the basis of what were selected the images.

Page 7 line 240: di-4-ANEPPS is known to internalize rapidly. Did they compare the results with a more lypophylic voltage dye (i.e di-8-ANEPPS)?

Supplementary Movie S1: the authors must indicate the distance between each frame. A scale bar is required.
Supplementary Movies S2-S11 time between each frame. Scale bars are missing.

Reviewer 2 Report

This manuscript describes a method to establish 3D culture of hippocampal neurons and fluorescence imaging of voltage and Ca ions using lattice lightsheet microscopy (LLSM). The LLSM is a high-speed volumetric imaging methods developed by the authors' group previously and the manuscript successfully demonstrated an application : realtime imaging of 3D cultured live neurons in gel, with voltage and chemical stimulation.

The reviewer believes the demonstration was successful and showed the advantages of their methods including LLSM. However, the information provided in the manuscript is somewhat insufficient for either understanding the principle, typical implementation, and the technical advantages of LLSM systems, establishing 3D culture in microsystems in the scope of the journal, and imaging of such systems. In other words, the reviewer did not fully catch the target readers the authors intended.

Therefore, the reviewer recommends adding more information on the technical background and key features of LLSM, extensive comparison between LLSM and other microscopy, and/or viewpoints on combining the 3D culture and the imaging system with microfluidic systems or other microsystems.

Minor comments:

P2L65 & P2L67

The authors may want to add names, model # or other detailed information of media used.

P2L66

The size and the method to form the 3D gel should be described here. The reviewer noticed this information appeared later and it seems to be too late.

P2L67

The 104 should be 104?

P2L70

The reviewer recommends adding more info on glial conditioning.

Figure 1.

This figure should be appeared in the Results section and it can be supplemental. In addition, the microscopy used should be stated. The reviewer recommends adding discussion on the image quality and other comparison of the methods used to take this figure with LLSM.

P3L19

The reviewer suggests adding the model # of the inverted research microscope system from Leica.

P3L100

The reviewer did not catch well how samples are set in the microscope systems. The author should clarify the "samples" mean just a gel sheet, a petri dish, or any other container. In addition, the "water" that immerses the samples can or should be cell culture medium, or other buffers?

Figure 2.

The panel A~D are just not understandable at least for the reviewer and thus did not help understand the system and the setup including the samples very well. The reviewer recommends addition of leaders, grouping, text information etc. Using illustration, mechanical 2D) drawings, or other 2D like views rather than solid bird views of an assembly in 3D CAD might be more helpful. The author cannot assume the readers read the authors' previous papers and they understand LLSM well.

Figure 2 captions

The reviewer recommends adding the title to the panel B, C, and D.

P5L116

Hank's should be "Hanks'"

P5L131

The word BNC is usually used as is like DNA, without showing "Bayonet-..."

P6L167

The reviewer recommends the authors adds more info on how the thickness of the gel droplet is maintained to 2 mm.

P6L192

There is no description on the "liquid-filled chamber" in this manuscript, and therefore the reviewer did not fully understand this statement well.

P6L195

The reviewer had several questions: what is the "imaging buffer"? How it can be separated from cell culture medium that should be with live cells? How to prevent contamination of cell culture during imaging?

P8L247

The duration and the interval should be clarified.

P8L250

The author may want to rephrase this sentence "ROIs (Bb) and neurons (Bc) is shown in (Bc)". The reviewer did not understand it.

Reviewer 3 Report

In this manuscript Chen et. al. investigate the use of Lattice Lightsheet Imaging for 3D culture systems in the context of organ on a chip technologies. The authors point out that 3D imaging at high spatiotemporal resolution is difficult with conventional spinning disk confocal microscopy due to light scattering and loss in 3D cultures. Instead LLSM is a more well suited approach. I enjoyed reading the manuscript which was well written and comprehensive. I feel that this paper provides a useful contribution to the field. I have a few minor points:

The authors refer to the image exposure of 6ms. Is this the actual frame rate? Similarly the pixel size is 0.102um, but the authors might comment on the actual resolution in the methods? Figure 2 A-D needs some labelling, the authors should not expect a non-expert to know what all of the components in the drawings represent. I didn't see an acutal characterisation of bleaching with the imaging parameters that were used. Can the authors comment in the methods how much bleaching there with their typical timelapse conditions. This should be discussed as it is a major motivation for using the technique. Some of the supplementary movies seemed to have a certain amount of vibration. Does this arise from the new low volume imaging paradigm?

Round 2

Reviewer 2 Report

The reviewer thinks appropriate revisions according to the most of comments the reviewers made. However, the reviewer found that Fig.2 C & D seems to be unchanged and still had to keep the same comments on them.

Reviewer 3 Report

The authors have addressed all of my comments.